# Quantitative Aspect of *Bacillus subtilis* σB Regulatory Network—A Computational Simulation

**DOI:** 10.3390/biology11121729

**Published:** 2022-11-29

**Authors:** Jiri Vohradsky

**Affiliations:** Laboratory of Bioinformatics, Institute of Microbiology, Czech Academy of Sciences, Vídeňská 1083, 142 20 Prague, Czech Republic; vohr@biomed.cas.cz

**Keywords:** *Bacillus subtilis* sigma B, computer simulation, regulatory network

## Abstract

**Simple Summary:**

A kinetic, ordinary differential equation model of regulation of sigma factor σB of *Bacillus subtilis* is presented. The model was derived from chemical equations describing interactions among σB, its anti-sigma and anti-anti-sigma factors RsbW, RsbV and phosphatases RsbU, RsbP which transmit environmental signals. The model uses experimental time series of gene expression which allowed to model the response of the system to the changes of its parameters using real expression data.

**Abstract:**

*Bacillus subtilis* is a model organism used to study molecular processes in prokaryotic cells. Sigma factor B, which associates with RNA polymerase, is one of the transcriptional regulators involved in the cell’s response to environmental stress. This study addresses the key question of how the levels of free SigB, which acts as the actual regulator of gene expression, are controlled. A set of chemical equations describing the network controlling the levels of free SigB was designed, leading to a set of differential equations quantifying the dynamics of the network. Utilizing a microarray-measured gene expression time series then allowed the simulation of the kinetic behavior of the network in real conditions and investigation of the role of phosphatases RsbU/RsbP transmitting the environmental signal and controlling the amounts of free SigB. Moreover, the role of kinetic constants controlling the formation of the molecular complexes, which consequently influence the amount of free SigB, was investigated. The simulation showed that although the total amount of sigma B is relatively high in the unstressed population, the amount of free SigB, which actually controls its regulon, is quite low. The simulation also allowed determination of the proportion of all the network members that were free or bound in complexes. While previously the qualitative features of *B. subtilis* SigB have been studied in detail, the kinetics of the network have mostly been ignored. In summary, the computational results based on experimental data provide a quantitative insight into the functioning of the SigB-dependent circuit and provide a roadmap for its further exploration in this industrially important bacterium.

## 1. Introduction

*Bacillus subtilis* is a Gram-positive bacterium with industrial applications for the production of antibiotics (bacitracin) and enzymes (amylases, lipases, proteases). Equally important is its exploration as a model organism to study fundamental bacterial molecular biology. *B. subtilis,* in order to survive in harmful conditions, forms spores [1]. When conditions improve, spores germinate and grow out [2]. These are complex processes that involve a number of regulatory factors, especially sigma factors. Sigma factors are critical for the recognition of promoters (DNA sequences where gene transcription starts). They bind RNA polymerase (RNAP) and form a holoenzyme. The holoenzyme, by means of the sigma factor, binds the specific promoter sequence and initiates the transcription of downstream genes. *B. subtilis* possesses 19 sigma factors of which SigA is the primary sigma factor that regulates the gene expression of housekeeping genes in the exponential phase [3,4].

Sigma B (SigB) is a principal stress response sigma factor protecting cells from various stressors such as heat, oxidative, alkaline, acidic or osmotic [5]. SigB recognizes two sequences in −35 and −10 (numbering relative to the transcription start site) regions GTTTaa and GGG(A/T)A(A/T) [6]. SigB itself is regulated by a mechanism involving anti-sigma, anti-anti-sigma factors and phosphatases that act upstream of the “anti” factors [7]. Under regular (stress-free) growth conditions, SigB is sequestered in the majority of cells [8] in a complex with its anti-sigma factor RsbW, which keeps it inactive. When the cell detects stress, the phosphatases RsbP and/or RsbU dephosphorylate RsbV that subsequently reacts with RsbW. SigB is released from the complex with its anti-sigma factor RsbW and activates its regulon, including its own operon, which also includes genes for RsbW and RsbV [8]. These interactions form a regulatory network whose activity was previously modeled by a set of chemical equations and rate constants [9], which explained some of the features observed in the experimental data. A kinetic analysis of the SigB regulon was presented in our previous study [6].

Although SigB is expressed under stress conditions, a comprehensive microarray-based study revealed that it is also expressed during the exponential phase without stress [10]. This study represents the so far most comprehensive measurement of *B. subtilis* gene expression, monitoring the germination and outgrowth of *B. subtilis* spores into the exponential phase. The dataset consists of 14 time points spaced by 5–10 min time intervals for 4008 genes.

SigB is present in the cell in several forms and chemical equilibria with its anti- and anti-anti-factors are experimentally (on the level of gene expression measurements) indistinguishable. In order to discover how SigB regulates its target genes, we have to know the levels of free SigB. In this study, I present a method using the experimentally determined gene expression time series [10] and an ordinary differential equation model derived from the chemical equations describing the SigB regulatory network to determine instant amounts of the molecules forming the network, either free or bound in complexes. This allowed analysis of the pivotal influence of phosphatases RsbP/RsbU on the amount of free SigB. It also allowed analysis of the influence of the kinetic constants controlling the crucial step of binding of the anti-sigma factor to SigB. The advantage of the presented approach is the use of real expression time series that allowed the computation of a level of free SigB, which actually acts as a regulator, for the particular experiment reflected in the time series. The presented model is a case study that demonstrates the possibility of using such an approach to model gene expression.

## 2. Materials and Methods

### 2.1. Data Acquisition

#### Time Series of Gene Expression

The *B. subtilis* transcriptomic microarray data were downloaded from GEO (http://www.ncbi.nlm.nih.gov/geo/query/acc.cgi?acc=GSE6865 (accessed on 20 July 2020)) consisting in 14 time points (0, 5, 10, 15, 20, 25, 30, 40, 50, 60, 70, 80, 90 and 100 min) during germination and outgrowth [10]. The dataset contains time series of expression of 4008 genes. Briefly (the experimental details can be found in the corresponding study [10]), 168 spores of *Bacillus subtilis* were generated by growing cells in a defined MOPS medium at 37 °C and shaking for 4 days. Spores were then thermally activated at 70 °C for 30 min in germination medium. The heat induction used for synchronization of the culture did not influence expression of SigB later on. The release of dipicolinic acid in the medium during spore germination was monitored using the terbium fluorescence assay. Samples for RNA isolation were drawn at regular intervals during germination and outgrowth. RNA was isolated from spores and outgrowing spores and Cy-labeled cDNA was produced by reverse transcription using Cy-labeled dUTP. Samples were hybridized to microarray slides; microarrays were scanned using an Agilent G2505 scanner. The experiment was carried out in duplicate (biological duplicate) and each sample was hybridized in duplicate on the microarray (technical duplicate). Data from replicates were averaged, and the original log2-based data were exponentiated.

Time series of gene expression of SigB, RsbW, RsbV, RsbP and RsbU were extracted from the abovementioned GEO database and used in the following paragraphs. Since the expression data are noisy, the data were interpolated using Matlab function interp1 (method “pchip”) at 1 min time steps and smoothed (Matlab function smoothdata, with a Gaussian-weighted moving average filter—method “gaussian”) prior to computation. The smoothing achieved more robust results with respect to high frequency phenomena expected in gene expression data, while preserving characteristic low frequency one. The expression profiles used for the computations are available in Appendix A.The system of differential equations was solved using Matlab ode23 solver with initial values set to zero.

## 3. Results and Discussion

The SigB regulatory scheme consists of several steps. Briefly, the anti-anti-sigma factor RsbV is dephosphorylated by two phosphatases, RsbP and RsbU. Dephosphorylated RsbV binds to the RsbW_2_ anti-sigma factor dimer to release SigB. RsbW also has a protein kinase activity that phosphorylates and inactivates RsbV (the complete scheme is a little bit more complicated and is given by the chemical Equations (ch1)–(ch8) and Figure 1).

A model describing the regulatory network controlling the activity of SigB was created. The aim was to determine how transcriptional and posttranscriptional events contribute to the regulation of SigB and the genes controlled by SigB and determine the actual amounts of molecular complexes and free forms of the sigma factor and its anti-factors. The SigB network is formally described by a set of biochemical Equations (ch1)–(ch8) [9] and the model is described by differential Equations (a1)–(a10) derived from the chemical equations.

The following equations symbolically describe interactions among the molecules of the network. Anti-sigma factor RsbW readily dimerizes and acts in the biochemical reactions as a dimer [11].

Binding of anti-sigma RsbW_2_ and anti-anti-sigma RsbV


(ch1)
$$RsbW_{2}+RsbV\begin{array}{c} k_{1+}\\ ⇄\\ k_{1-} \end{array}RsbW_{2}RsbV$$



(ch2)
$$RsbW_{2}RsbV+RsbV\begin{array}{c} k_{2+}\\ ⇄\\ k_{2-} \end{array}RsbW_{2}RsbV_{2}$$


2.Phosphorylation of anti-anti-sigma factor by anti-sigma factor


(ch3)
$$RsbW_{2}+RsbV+P\overset{k_{3+}}{\to }RsbW_{2}+RsbV-P$$



(ch4)
$$RsbW_{2}RsbV_{2}+P\overset{k_{4+}}{\to }RsbW_{2}RsbV+RsbV-P$$


3.Binding of sigma factor to anti-sigma factor


(ch5)
$$RsbW_{2}+SigB\begin{array}{c} k_{5+}\\ ⇄\\ k_{5-} \end{array}RsbW_{2}SigB$$


4.Displacement of SigB by RsbV in complex with RsbW_2_


(ch6)
$$RsbW_{2}SigB+RsbV\begin{array}{c} k_{6+}\\ ⇄\\ k_{6-} \end{array}RsbW_{2}RsbV+SigB$$


5.Dephosphorylation of anti-anti-sigma factor by RsbU RsbP


(ch7)
$$RsbVP+RsbU\overset{k_{7+}}{\to }RsbV+RsbU+P$$



(ch8)
$$RsbVP+RsbP\overset{k_{7+}}{\to }RsbV+RsbP+P$$


The measured values of SigB, RsbV and RsbW were considered as composed of the free forms together with the forms bound in the complexes (Equations (a1)–(a3)) as given by the chemical equations above. The measured expression profile includes both synthesis and degradation that, therefore, does not have to be considered explicitly. In the next equation, the following abbreviated notation will be used: V = RsbV, W = RsbW, S = measured total amount of SigB, AS = measured total amount of RsbW, AAS = measured total amount of RsbV. Index *f* = free form of the factor. Due to the linearity of translation, the mRNA expression profiles can be used in the simulation instead of the protein ones.
(a1)$$\left[S\right]=[SigB_{f}]+[W_{2}SigB]$$
(a2)$$\left[AS\right]=2[W_{2f}]+2[W_{2}V]+2[W_{2}V_{2}]+2[W_{2}SigB]$$
(a3)$$\left[AAS\right]=[V_{f}]+[W_{2}V]+2\left[W_{2}V_{2}\right]+[VP]$$

By differentiating Equations (a1)–(a3) and the kinetic equations of ch1–ch8 we obtain
(a4)$$\frac{d[SigB_{f}]}{dt}=\frac{d\left[S\right]}{dt}-\frac{d\left[W_{2}SigB\right]}{dt}$$
(a5)$$\frac{d[W_{2f}]}{dt}=\frac{1}{2}\frac{d\left[AS\right]}{dt}-\frac{d[W_{2}V]}{dt}-\frac{d[W_{2}V_{2}]}{dt}-\frac{d[W_{2}SigB]}{dt}$$
(a6)$$\frac{d[V_{f}]}{dt}=\frac{d\left[AAS\right]}{dt}-\frac{d\left[W_{2}V\right]}{dt}-2\frac{d\left[W_{2}V_{2}\right]}{dt}-\frac{d\left[VP\right]}{dt}$$
(a7)$$\frac{d\left[W_{2}SigB\right]}{dt}=k_{5+}[W_{2}]\left[SigB_{f}\right]-k_{5-}\left[W_{2}SigB\right]-k_{6+}\left[W_{2}SigB\right][V_{f}]+k_{6-}[W_{2}V]\left[SigB_{f}\right]$$
(a8)$$\begin{array}{l} \frac{d[W_{2}V]}{dt}=k_{2-}[W_{2}V_{2}]-k_{2+}\left[W_{2}V\right][V_{f}]+k_{4+}[W_{2}V_{2}]+k_{1+}\left[W_{2f}\right][V_{f}]-k_{1-}[W_{2}V]+k_{6+}\left[W_{2}SigB\right]\left[V_{f}\right]\\ \;\;\;\;\;\;\;\;\;\;\;\;\;\;\;\;\;\;\;\;\;\;\;\;\;\;-k_{6-}\left[W_{2}V\right]\left[SigB_{f}\right] \end{array}$$
(a9)$$\frac{d[W_{2}V_{2}]}{dt}=k_{2+}\left[W_{2}V\right]\left[V_{f}\right]-k_{2-}\left[W_{2}V_{2}\right]-k_{4+}\left[W_{2}V_{2}\right]$$
(a10)$$\frac{d\left[VP\right]}{dt}=k_{3+}[W_{2f}]\left[V_{f}\right]+k_{4+}[W_{2}V_{2}]-k_{7+}\left[VP\right]\left[RsbP\right]-k_{7+}\left[VP\right]\left[RsbU\right]$$
where the constants k_i_ are given in Table 1.

Expression profiles and their derivatives of SigB, RsbW, RsbV (S, AS and AAS), RsbP and RsbU are readily available, and their expression profiles are shown in Figure 2 and are available in Appendix A. The Equations (a4)–(a10) were solved numerically in Matlab.

Figure 2 shows marked differences in the expression levels of SigB and its anti- and anti-anti-factors compared to the expression levels of the phosphatases. While the SigB operon genes (*sigB*, *rsbV*, *rsbW*) reached values >10,000, the phosphatases levels were much lower, only ~1000 (*RsbP*) and ~2500 (*RsbU*). As the phosphatases control the amounts of free anti-anti-sigma factor, it should also be reflected in the amounts of free SigB. In order to determine the amounts of free SigB and other individual molecules and complexes of the network, a simulation solving Equations (a4)–(a10) was performed. The simulation was conducted using the smoothed expression profiles from Figure 2 and the constants from Table 1, the results are shown in Figure 3.

Figure 3 shows that most of the anti-anti-sigma factor RsbV remained phosphorylated (compare with the dashed line that shows total RsbV). In consequence, the anti-sigma factor RsbW bound in the two complexes with RsbV remained low and SigB bound by the anti-sigma factor was relatively high. This resulted in a relatively low amount of free SigB (sigB_f_). Keijser et al. [10], in their paper, mentioned surprisingly high levels of SigB that were found in their non-stressed culture. The simulation presented here shows that although the total amount of SigB appears to be higher than expected, the free form of SigB, which acts as a regulator, was almost four times lower, which is quite natural for non-stressed conditions.

### 3.1. RsbU/RsbP Influence

To investigate the influence of RsbU/RsbP, a simulation of the SigB regulatory network for increasing expression levels of RsbU/RsbP was performed. The results are summarized in Figure 4.

The simulation showed that an increase in RsbU/RsbP expression resulted in a marked decrease in the level of phosphorylated RsbV (RsbVP), which subsequently caused an increase in the expression of free RsbV that can then actively block the anti- sigma factor. Consequently, the amount of free SigB increased. It is also apparent that when the increase in RsbU/RsbP was 5 to 10 fold, the increase in free SigB was lower, indicating the importance of the rate constants controlling the binding of RsbW_2_ to SigB; a relatively high amount of SigB still remains bound in the complex with RsbW. The simulation of the RsbU/RsbP influence thus showed that the posttranslational regulation of SigB keeps free SigB within certain limits as dictated by the expression of RsbU and RsbP and the corresponding rate constants, which are controlled externally. To investigate the influence of rate constants, I simulated different values of the rate constants controlling the binding of RsbW_2_ to SigB (Equation (ch5)) and the displacement of SigB by RsbV in the complex with RsbW_2_ (Equation (ch6), see next paragraph).

### 3.2. Influence of Rate Constants

There are two rate limiting steps in the chemical equations describing the SigB regulating network: the binding of anti-sigma factor to SigB (Equation (ch5)), and the displacement of the sigma factor by the anti-anti-sigma factor (Equation (ch6)). The amounts of free and/or bound components are controlled by the rate constants’ estimates as specified in Table 1. I stress here that the values of these constants may somewhat differ from the reality (they were an approximation based on available data). In the next two paragraphs, the abovementioned rate constant ratios were varied in order to investigate their influence on the expression of the network members.

#### 3.2.1. K_5−_/k_5+_ Ratio

The original ratio of k_5-_/k_5+_ given by Table 1 was 5000. Figure 5 then shows variations of this ratio from 100 to 50,000 to obtain a high scatter. Figure 5 illustrates that an increase in the k_5-_/k_5+_ ratio led to an increase in free SigB so that it was almost the only form of SigB. As expected, this was accompanied by a decrease in the complex of SigB with its anti-sigma factor and by an increase in free RsbW_2f_. The complexes of anti- and anti-anti-sigma factors were affected only moderately as they are controlled by different kinetic constants.

#### 3.2.2. K_6+_/k_6−_ Ratio

Constants k_6+_/k_6−_ controlling displacement of sigma factor by anti-anti-sigma factor (Equation (ch6)) were varied in the interval <1 to 100,000>. The results are shown in Figure 6.

Apparently, shifting the reaction in the Equation (ch6) to the left by increasing the ratio k_6−_/k_6+_ caused an increase in the amounts of free SigB and a decrease in the SigBRsbW_2_ complex. The amounts of free anti-factors behaved in the opposite way: RsbV_f_ decreased, while RsbW_2f_ increased. A large portion of RsbV remained phosphorylated as the amounts of the phosphatases were not changed and were small.

## 4. Conclusions

The results presented in this study reveal the dynamics of SigB, showing how much of SigB is bound in complexes and how much of it is available for the regulation of other genes from its regulon. It shows that under the regular conditions of growth recorded in the gene expression data, most of the SigB is bound in a complex with its anti-sigma factor, leaving just a small portion available for further activity. The simulation showed that its free form is minor. The reason is that due to exponential regular growth in rich media, free SigB is not necessary and is released from the complex only by an external signal controlling the phosphatases RsbU/P. The simulations show that the stress sigma factor is not necessarily only expressed under stress (or in developmental states of the cell requiring, e.g., swimming/swarming [12]) but can already be present, waiting for the signal.

Analysis of the influence of kinetic constants revealed that most important were the constants controlling the binding of anti-sigma factor to SigB and these constants also influenced the sensitivity of the control by means of the phosphatases dephosphorylating anti-anti-sigma factor. As the constants that I used here are mostly semi-quantitative estimates, the actual limits of free SigB given by them are, therefore, only indicative. More important are the trends and sensitivities of the levels of the network members to the changes in these parameters.

Last but not least, it should be mentioned that the simulations were carried out with mRNA expression levels instead of the protein ones. The argument for the use of mRNA profiles is that the translation process is linear with respect to time and the size of mRNA and is several orders faster than the changes observed during the development of the population (posttranscriptional control has not been reported for this case). The argument against using mRNA levels instead of the protein levels is that some posttranslational regulation may occur. In the model described here, such posttranslational events that actually control the amounts of free SigB were fully incorporated. In any case, using the protein expression instead of mRNA may change the final amounts of protein, but, due to the linearity of translation, it would not influence the expression profile that is most important for the simulation; it may influence the final amounts of the products of the network, but the principles and the results will remain the same.

In summary, the analysis of the SigB regulatory network presented here provides novel insights into the quantitative principles of the regulation of a group of sigma factors to which SigB belongs. It provides a blueprint for the discovery of quantitative principles for factors whose qualitative aspects have already been established.

## Figures and Tables

**Figure 1 biology-11-01729-f001:**
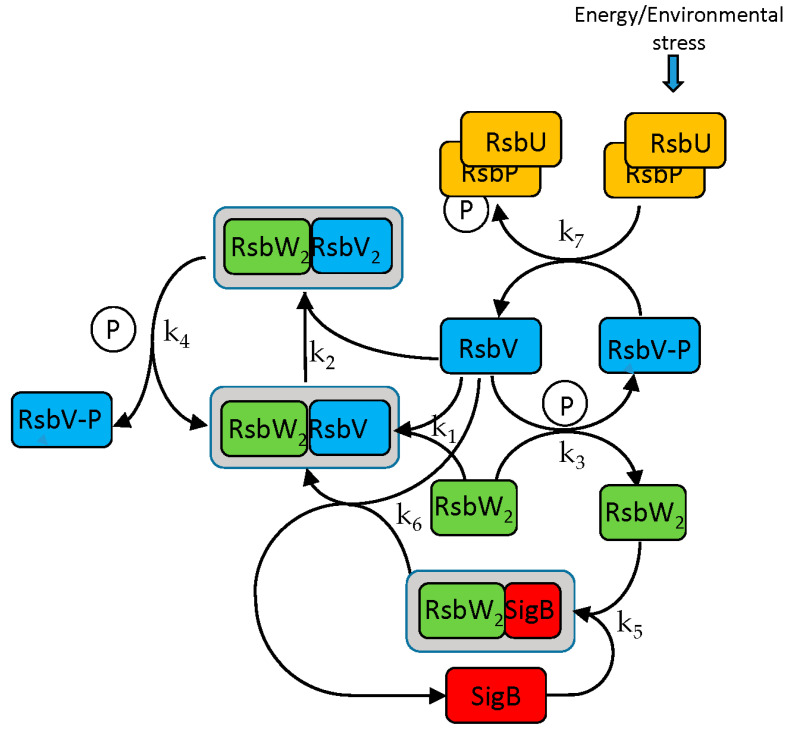
SigB regulatory network. Anti-anti-sigma factor RsbV is dephosphorylated by two phosphatases, RsbP and RsbU. Dephosphorylated RsbV binds to the RsbW_2_ anti-sigma factor dimer to release SigB bound in a complex with RsbW_2_. RsbW also has a protein kinase activity that phosphorylates and inactivates RsbV. **P** represents phosphate. Complete description in the form of chemical Equations (ch1)–(ch8) is given below. k_1_–k_7_ are corresponding kinetic constants. Grey frame represent complexes.

**Figure 2 biology-11-01729-f002:**
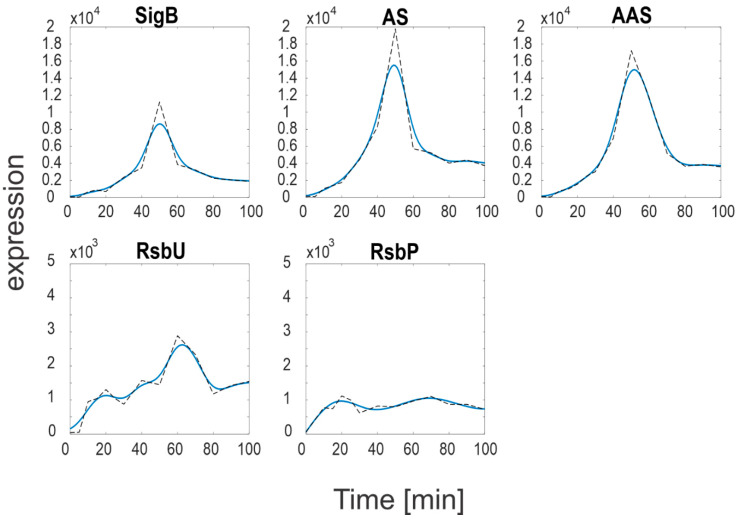
Measured (dashed line) and approximated (solid line) expression profiles of genes of the SigB regulatory network. S-measured total amount of SigB, AS-measured total amount of RsbW, AAS-measured total amount of RsbV. Note a different scale for RsbU and RsbP that is two orders of magnitude lower than that of the SigB operon.

**Figure 3 biology-11-01729-f003:**
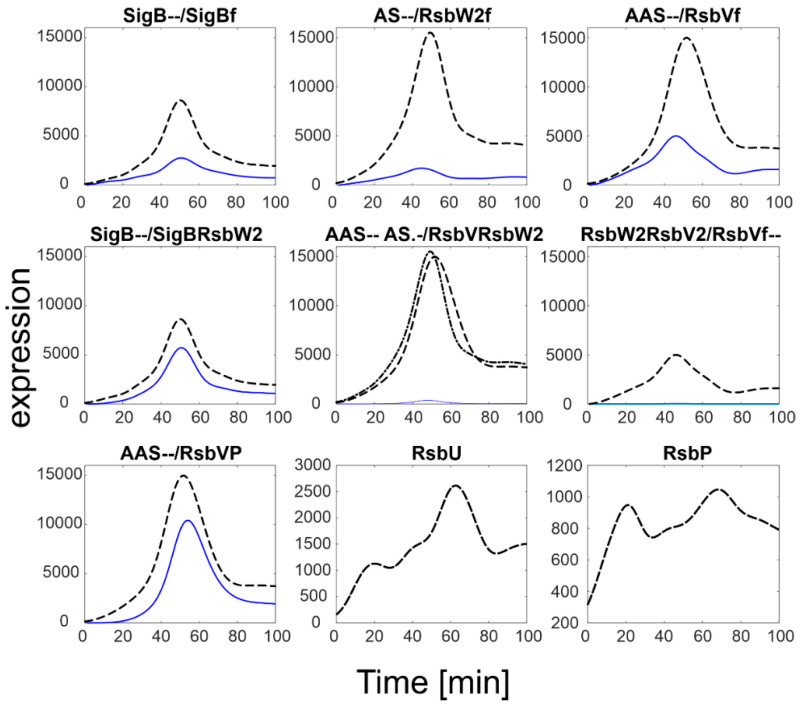
SigB regulatory network simulation results using default parameters from Table 1. Black dashed lines represent smoothed expression profiles of the total amount of the given molecule as measured experimentally, blue solid lines the modeling results. S-measured total amount of SigB, AS-measured total amount of RsbW, AAS-measured total amount of RsbV, index f-free form of the factor. Note a different scale for RsbU and RsbP that is two orders of magnitude lower than that of the SigB operon. Note the scales of RsbU/P are much lower than that of the others.

**Figure 4 biology-11-01729-f004:**
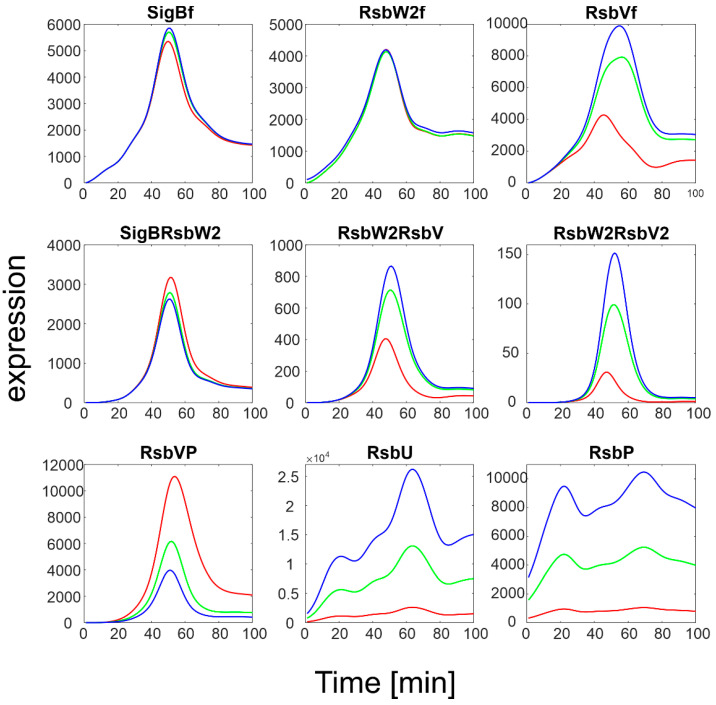
Results of simulation of SigB regulatory network with increasing expression of RsbU/RsbP. Red line represents original measured values of RsbU/RsbP and the resulting expression of the remaining members of the network. Index f-free form of the factor. Green and blue lines are 5 and 10 multiples of the red line for RsbU/RsbP, which served as input variables, respectively.

**Figure 5 biology-11-01729-f005:**
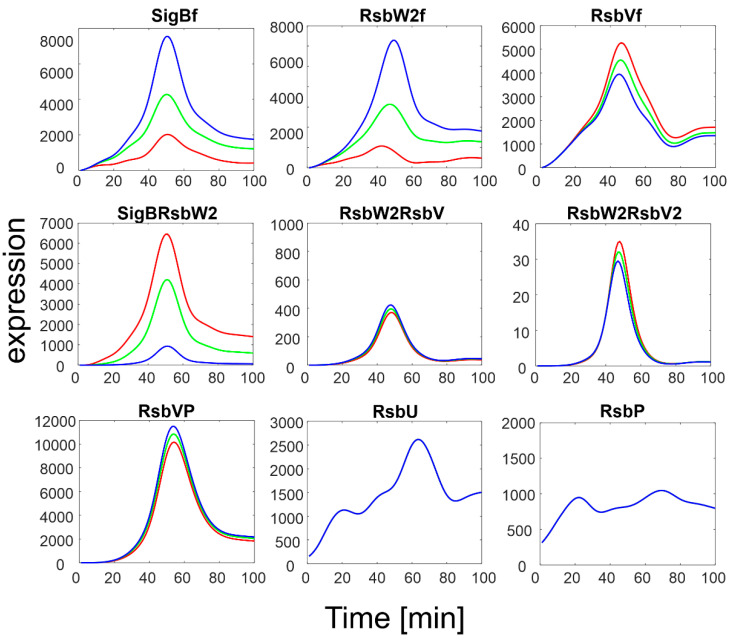
Simulation of the influence of the k_5−_/k_5+_ ratio on the binding of anti-sigma factor to SigB. Red- k_5−_/k_5+_ = 100, green, k_5−_/k_5+_ = 2.5.10^3^, blue k_5−_/k_5+_ = 5.10^4^. The original ratio was 5.10^3^. Index f-free form of the factor.

**Figure 6 biology-11-01729-f006:**
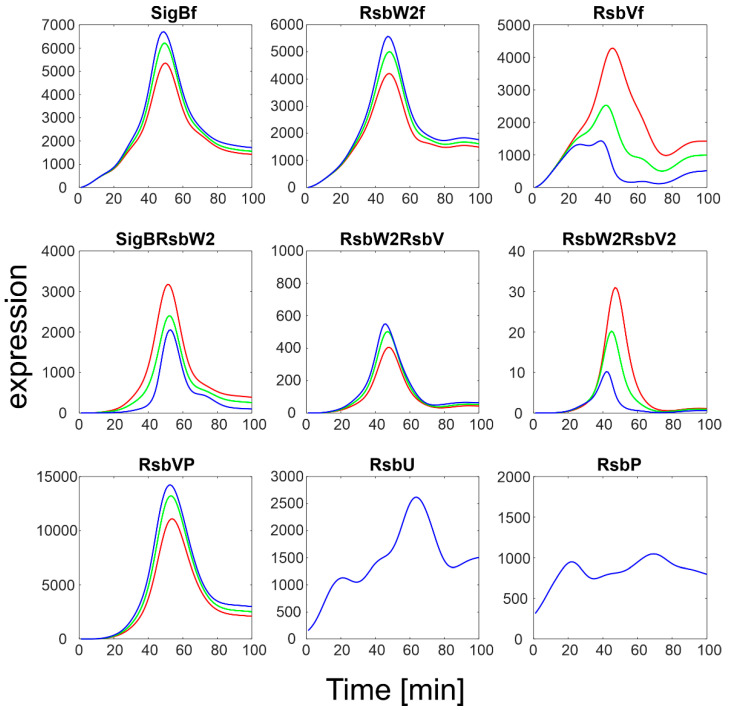
Simulation of influence of the ratio k_6−_/k_6+_ controlling displacement of sigma factor by anti-anti-sigma factor. Red: original values as in Table 1, green: k_6−_/k_6+_ = 100, blue: k_6−_/k_6+_ = 1x10^5^.

**Table 1 biology-11-01729-t001:** Kinetic constants [9] scaled and converted to min^−1^ to fit microarray-measured expression profiles (y).

Rate Constant	Value
k_1+_	6.10^−5^ y^−1^ min^−1^
k_1−_	0.3 min^−1^
k_2+_	6.10^−5^ y^−1^ min^−1^
k_2−_	0.3 min^−1^
k_3+_	3 min^−1^
k_4+_	3 min^−1^
k_5+_	6.10^−5^ y^−1^ min^−1^
k_5−_	0.3 min^−1^
k_6+_	3.10^−5^ y^−1^ min^−1^
k_6−_	3.10^−5^ y^−1^ min^−1^
k_7+_	3.10^−5^ y^−1^ min^−1^

## Data Availability

*B. subtilis* transcriptomic microarray data were downloaded from GEO (http://www.ncbi.nlm.nih.gov/geo/query/acc.cgi?acc=GSE6865 (accessed on 20 July 2020)).

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
