# Peer review of "Quantitative Aspect of Bacillus subtilis σB Regulatory Network—A Computational Simulation"

_biology, 2022, doi:10.3390/biology11121729_

Round 1
Reviewer 1 Report (New Reviewer)
In the manuscript, Quantitative aspect of Bacillus subtilis σB regulatory network, the author presents a simulation approach to describe the sigma B regulatory network.
The authors use available transcriptomics data and previously determined kinetic constants to develop the simulation model. The report is interesting as it explores a classical complex stress regulation pathway. Also, the simulation data is based on transcriptomics; the report also describes the complex regulatory interplay between Sigma B and its cognate anti and anti-anti sigma factor. The conclusion describes a stress response system that capitalizes on already present Sigma B protein.
I believe the report is of general interest to the Biology reader and could be presented with minor modifications.
In the result and discussion, capturing all the abbreviations used in the formulas is difficult. I would suggest presenting a summary diagram depicting the protein interactions, dimerization, and phosphorylation.
Equation (2) depicting the Phosphorylation reaction was shifted in the PDF.
For the reaction (ch3). Shouldn’t the RsbV phosphorylated appear as RsbV-P?
In equations (a1) and (a2), both terms SigBW2 and W2SigB are used; this is confusing for the reader.
The legend of Fig.1 and Fig.2 should recapitulate the signification of the abbreviation (S, AS, AAS).
In the conclusion
Lane 243, a bracket is missing.
Also, in conclusion, the author explains that because of the “linearity of the translation” process, it is not essential to integrate protein level to elaborate the Sigma B regulatory network. The statement is OK but shouldn’t it be presented earlier in the manuscript (result and discussion)?
Author Response
In the manuscript, Quantitative aspect of Bacillus subtilis σB regulatory network, the author presents a simulation approach to describe the sigma B regulatory network.
The authors use available transcriptomics data and previously determined kinetic constants to develop the simulation model. The report is interesting as it explores a classical complex stress regulation pathway. Also, the simulation data is based on transcriptomics; the report also describes the complex regulatory interplay between Sigma B and its cognate anti and anti-anti sigma factor. The conclusion describes a stress response system that capitalizes on already present Sigma B protein.
I believe the report is of general interest to the Biology reader and could be presented with minor modifications.
In the result and discussion, capturing all the abbreviations used in the formulas is difficult. I would suggest presenting a summary diagram depicting the protein interactions, dimerization, and phosphorylation.
RE: A new figure (Figure 1) was added describing schematically reactions given in the equations ch1-ch8. The other figures were renumbered.
Equation (2) depicting the Phosphorylation reaction was shifted in the PDF.
For the reaction (ch3). Shouldn’t the RsbV phosphorylated appear as RsbV-P?
RE: Errors during conversion. Changes to RsbV were made.
In equations (a1) and (a2), both terms SigBW2 and W2SigB are used; this is confusing for the reader.
RE: Thanks. SigBW2 was changed to W2SigB consistently with the rest of the equations.
The legend of Fig.1 and Fig.2 should recapitulate the signification of the abbreviation (S, AS, AAS).
RE: the legend was changed to (adding the new figure 1 numbering of figures changed): “Figure 2. Measured (dashed line) and approximated (solid line) expression profiles of genes of the SigB regulatory network. S-measured total amount of SigB, AS-measured total amount of RsbW, AAS-measured total amount of RsbV. Note a different scale for RsbU and RsbP that is two orders of magnitude lower than that of the SigB operon.”
“Figure 3. SigB regulatory network simulation results using default parameters from Table 1. Black dashed lines represent smoothed expression profiles of the total amount of the given molecule as measured experimentally, blue solid lines the modeling results. S-measured total amount of SigB, AS-measured total amount of RsbW, AAS-measured total amount of RsbV, index f-free form of the factor. Note a different scale for RsbU and RsbP that is two orders of magnitude lower than that of the SigB operon. Note the scales of RsbU/P are much lower than that of the others.”
In the conclusion
Lane 243, a bracket is missing.
RE: change was made.
Also, in conclusion, the author explains that because of the “linearity of the translation” process, it is not essential to integrate protein level to elaborate the Sigma B regulatory network. The statement is OK but shouldn’t it be presented earlier in the manuscript (result and discussion)?
RE: a sentence “due to the linearity of translation, the mRNA expression profiles can be used in the simulation instead of the protein ones” was added to the line 127.

Reviewer 2 Report (New Reviewer)
The author analyze the free SigB in B. subtilis via a model based on a microarray measured gene expression time series. However, the significance of the model was not demonstrated, and the readability in the manuscript was insufficient for readers.
1. Considering of the model was derived from a reported SigB network biochemical equations, the significance of the model and method for determination of free SigB in B. subtilis should be clarified.
2. The biochemical equations (ch1- ch8) were blurry.
3. Figure legends in the manuscript (especially Fig 2 and Fig 4) should be enriched, some of the description of the figure title (SigBf, RsbW2f,...) should be added in the lengend and make it easy to read.
4. In Figure 2, two patterns of black dashed line were appeared without description. In Figure 3, the title of figures in top panel were cutoff.
5. The discussion section of this article is too general and lacks in-depth analysis of the results.
Author Response
The author analyze the free SigB in B. subtilis via a model based on a microarray measured gene expression time series. However, the significance of the model was not demonstrated, and the readability in the manuscript was insufficient for readers.
- Considering of the model was derived from a reported SigB network biochemical equations, the significance of the model and method for determination of free SigB in B. subtilis should be clarified.
RE: The novelty of the simulation is in incorporating real measured expression profiles, which allowed simulation of behavior of the system under particular conditions and see how the system behave when the parameters of the model change. This is mentioned in abstract: “Utilizing a microarray measured gene expression time series allowed then to simulate kinetic behavior of the network in real conditions and investigate the role of phosphatases RsbU/RsbP transmitting the environmental signal and controlling the amounts of free SigB.”
in introduction: “I present a method using the experimentally determined gene expression time series [10] and an ordinary differential equation model derived from the chemical equations describing the SigB regulatory network to determine instant amounts of the molecules forming the network, either free or bound in complexes. This allowed to analyze the pivotal influence of phosphateses RsbP/RsbU on the amount of free SigB. It also allowed to analyze the influence of the kinetic constants controlling the crucial step of binding of the anti- sigma factor to SigB. The advantage of the presented approach is the use of real expression time series that allowed to compute free SigB (or any other similar factor with known regulatory network) level for the particular experiment reflected in the time series”.
And Conclusions: “It shows that under the regular conditions of growth recorded in the gene expression data…..”.
- The biochemical equations (ch1- ch8) were blurry.
RE: it is because the latex formulas were converted to gif. I can provide original latex formulas if the publisher allows to do that.
- Figure legends in the manuscript (especially Fig 2 and Fig 4) should be enriched, some of the description of the figure title (SigBf, RsbW2f,...) should be added in the lengend and make it easy to read.
RE: the changes were made already under reviewer 1 comments. Please note that a new Figure 1 was added and the numbering of figures therefore changed. The legends are:
Figure 3. SigB regulatory network simulation results using default parameters from Table 1. Black dashed lines represent smoothed expression profiles of the total amount of the given molecule as measured experimentally, blue solid lines the modeling results. . S-measured total amount of SigB, AS-measured total amount of RsbW, AAS-measured total amount of RsbV, index f-free form of the factor. Note a different scale for RsbU and RsbP that is two orders of magnitude lower than that of the SigB operon.Note the scales of RsbU/P are much lower than that of the others.
Figure 4. Results of simulation of SigB regulatory network with increasing expression of RsbU/RsbP. Red line represents original measured values of RsbU/RsbP and the resulting expression of the remaining members of the network. index f-free form of the factor. Green and blue lines are 5 and 10 multiples of the red line for RIbU/RsbP which served as input variables respectively.
Figure 5. Simulation of the influence of the k5-/k5+ ratio on the binding of anti- sigma factor to SigB. Red- k5-/k5+=100, green, k5-/k5+=2.5.103, blue k5-/k5+= 5.104. The original ratio was 5.103. Index f-free form of the factor.
- In Figure 2, two patterns of black dashed line were appeared without description. In Figure 3, the title of figures in top panel were cutoff.
RE: Figure 2. In the caption of figure 2 the black lines are labeled as AAS—and AS.- describing the meaning of the lines.
Figure3. error probably appeared during upload and conversion. Original figures are OK.
- The discussion section of this article is too general and lacks in-depth analysis of the results.
RE: The details are given in Results and Discussion. Conclusions are made to briefly summarize the results. So the in-depth analysis is given above. I am afraid that adding some broader discussion might lead to speculations for which I have no strong enough evidence, and which I would like to avoid. If specified in more details, I am ready to make changes. Btw. Reviewer 3 requires that the Conclusions are too broad and should be shortened. (?).

Reviewer 3 Report (New Reviewer)
The scientific name “Bacillus subtilis” in title should be italic, check all manuscript
Indicate abbreviations in line 11,15
Add some numerical data in abstract
Line 40 is not clear, check
Again, indicate abbreviation in line 44
Lines 57,58 check for clarity
Extend the introduction with some recent citation and detail the mechanism of Sigma factor
Clear the objectives at the end of introduction
Line 79 MOPS?
Make list of abbreviations at the end of manuscript
Line 87 add city, model of all devices used in manuscript
Add section materials for all exist materials in manuscript
Equation must be rewritten by MS equations and indicate “a”, and “ch” in all Equations
Clear the values in Table 1 “6.10-5 x-1min-1” is X as multiple or what
Rewrite titles 3.2.1 and 3.2.2 k5-/k5+ ration,
Enhance Figure 4, 3 enhance regarding labels
Reduce conclusion
Where the discussion part adds discussion for each part of manuscript with recent citations
Author Response
The scientific name “Bacillus subtilis” in title should be italic, check all manuscript
RE: manuscript was checked and changes were made
Indicate abbreviations in line 11,15
RE: I don’t understand. If you mean SigB, RsbV, RsbW, RsbU, RsbP they are standard names of genes. I am submitting the manuscript to Biology journal whose readers understand them.
--------
Add some numerical data in abstract
RE: please, which numerical data?
--------------
Line 40 is not clear, check
RE: the sentence was changed to :” SigB is released from the complex with its anti-sigma factor RsbW and activates its regulon, including its own operon, which also includes genes for RsbW and RsbV [8]”
-------------
Again, indicate abbreviation in line 44.
RE: please see answer to query 2.
----------------
Lines 57,58 check for clarity
RE: The sentence was changed to: The advantage of the presented approach is the use of real expression time series that allowed to compute a level of free SigB, which actually acts as a regulator, for the particular experiment reflected in the time series.
Extend the introduction with some recent citation and detail the mechanism of Sigma factor
RE: a paragraph: “. Sigma factors are critical for recognition of promoters (DNA sequences where gene transcription starts) are subunits of They bind RNA polymerase (RNAP) that are critical for recognition and form a holoenzyme of promoters (DNA sequences where gene transcription starts). The holoenzyme by means of the sigma factor binds specific promoter sequence and initiates transcription of downstream genes. B. subtilis possesses 19 sigma factors of which SigA is the primary sigma factor that regulates gene expression of housekeeping genes in exponential phase [3,4].”, was added starting at line 32.
I am not aware of any recent publications on the topic of SigB regulatory network.
------------
Clear the objectives at the end of introduction
Line 79 MOPS?
Make list of abbreviations at the end of manuscript
RE: the only abbreviations I use are S, AS, AAS, which are explained in the text and in the figure legends. I can make a list, but is it worth for 3 items?
------------------
Line 87 add city, model of all devices used in manuscript
RE: I haven’t used any particular devices except for a standard PC.
------------------
Add section materials for all exist materials in manuscript
RE:I used published data. No materials except for the standard PC were used.
-----------------
Equation must be rewritten by MS equations and indicate “a”, and “ch” in all Equations
RE: MS math editor does not allow to write chemical equations. The chemical equations were written in LaTex and inserted to the manuscript as gif. The LeTex formula are available and can be provided to the publisher.
-----------------
Clear the values in Table 1 “6.10-5 x-1min-1” is X as multiple or what
RE: X is not a multiple, it is represents the units in which the microarrays were measured. The title was changed to: “Kinetic constants [9] scaled and converted to min-1 to fit microarray measured expression profiles (y).”
------------------
Rewrite titles 3.2.1 and 3.2.2 k5-/k5+ ration,
RE: changes were made.
------------------
Enhance Figure 4, 3 enhance regarding labels
RE: labels are of the same font, size, and face for all figures. I don’t know which labels in particular should be enhanced and how?
---------------------
Reduce conclusion
RE: Reviewer 2 wants to expand Conclusions…. (?)
---------------------
Where the discussion part adds discussion for each part of manuscript with recent citations
RE: All relevant citations were listed. It is necessary to emphasize that the SigB circuit is known for long time and its study has not been updated for rather long time too. For this reason there are no recent publications on this topic.

Reviewer 4 Report (New Reviewer)
Dear editor,
I went throughout the whole manuscript carefully, and the manuscript is written in a very confusing way. English revision is abslutely necessary. The reference list is extremely short, showing that the authors did not study the field properly. I am concerned about the quality of these data and their imporntance to the field. I would reccoment not to publish this manuscript in biology journal.
Kind regards,
Agata OLszewska-Widdrat
Author Response
I went throughout the whole manuscript carefully, and the manuscript is written in a very confusing way. English revision is abslutely necessary. The reference list is extremely short, showing that the authors did not study the field properly. I am concerned about the quality of these data and their imporntance to the field. I would reccoment not to publish this manuscript in biology journal.
Kind regards,
Agata OLszewska-Widdrat
RE: concerning the references - I reported all which were used. It is necessary to emphasize that the SigB network has been known for long time and the work on it has not been updated. The recent publications can’t be referenced as they do not exist.
Reviewer 5 Report (New Reviewer)
In the current study, the author reported a computer simulation of the SigB regulatory network in Bacillus subtilis.
Please add how this study can be continued and which could be its practical applicability. The article could be more valuable if it had also a small experimental part of its own. It’s true that Biology Journal has as subject areas mathematical biology/ biomathematics/bioinformatics, so I consider that the article is suitable for this publication.Author Response
In the current study, the author reported a computer simulation of the SigB regulatory network in Bacillus subtilis.
Please add how this study can be continued and which could be its practical applicability. The article could be more valuable if it had also a small experimental part of its own. It’s true that Biology Journal has as subject areas mathematical biology/ biomathematics/bioinformatics, so I consider that the article is suitable for this publication.
RE: It’s a basic research topic, its aim is to elucidate particular molecular process in cell. I don’t see any practical applicability.
Round 2
Reviewer 2 Report (New Reviewer)
The author has made relevant changes in the article to address my previous issues and has also provided a schematic diagram of the regulation network of the sigB factor to improve the readability of the article.
Reviewer 3 Report (New Reviewer)
accepted
This manuscript is a resubmission of an earlier submission. The following is a list of the peer review reports and author responses from that submission.
Round 1
Reviewer 1 Report
-
The title is not understandable, please rephrase it. Eg: Modelling the quantitative aspect of Bacillus subtilus SigB regulatory network - a computational simulation” if I understand it correctly. The article “the” in the middle of the title is confusing.
-
What are Sigma factor B and SigB? Are they different? So what is the meaning of the sentence “The simulation showed that although the total amount of free sigma B is relatively high in the unstressed population, the amount of free SigB is quite low.” ?
-
“ … and showed that the amount of free SigB is controlled to be kept within particular limits.” not clear to me.
The abstract should be rewritten, and the results should be clearly stated.
Line 32: locus where gene transcription ….. Instead.
Material and method
I noticed that the author had used data from ref 10, similar to his previous publication, “Kinetic Modeling and Meta-Analysis of the Bacillus subtilis SigB Regulon during Spore Germination and Outgrowth”. Can the author explain the importance of this new finding and why it could not be discussed in the previous publication?
Line 79: what is RsbW, RsbV, RsbP and RsbU?
I don’t see why the author cites previous publications in his finding, is it a review? Can the author explain how he came to these equations?
It is difficult for me to figure out the difference between previous data and this work findings.
Eg: table 1: why does the author have to paste this table from ref 9 while the data were from ref 10? I am confused.
Author Response
The title is not understandable, please rephrase it. Eg: Modelling the quantitative aspect of Bacillus subtilus SigB regulatory network - a computational simulation” if I understand it correctly. The article “the” in the middle of the title is confusing.
RE: The title was misspelled. It was changed to “Quantitative aspect of Bacillus subtilis σB regulatory network - a computational simulation”.
2.
What are Sigma factor B and SigB? Are they different? So what is the meaning of the sentence “The simulation showed that although the total amount of free sigma B is relatively high in the unstressed population, the amount of free SigB is quite low.” ?
RE: SigB is standardly used abbreviation for Sigma B or sigma factor B.
The sentence was formulated wrong, it should be:” The simulation showed that although the total amount of sigma B is relatively high in the unstressed population, the amount of free SigB is quite low.”
“ … and showed that the amount of free SigB is controlled to be kept within particular limits.” not clear to me.
RE: The simulation in paragraph 3.1.RsbU/RsbP influence, and corresponding Figure 4 shows that changes in the phosphatases amounts controlling indirectly SigB hve limits in the effect to the amounts of free SigB. This sentence is confusing and was deleted.
The abstract should be rewritten, and the results should be clearly stated.
RE: changes to abstract were made.
Line 32: locus where gene transcription ….. Instead.
RE: changes were made
Material and method
I noticed that the author had used data from ref 10, similar to his previous publication, “Kinetic Modeling and Meta-Analysis of the Bacillus subtilis SigB Regulon during Spore Germination and Outgrowth”. Can the author explain the importance of this new finding and why it could not be discussed in the previous publication?
RE: The previous paper deals with SigB regulon, while this paper deals with the regulation of SigB itself, which are two different topics. It was not included in the paper because the results of this paper were not ready at the time of its publication. I find the importance of the findings in this paper in defining the kinetics of all components of the SigB regulating system, particularly for with utilization of measured data. This makes difference to all other previous analyses based on kinetic models, which used only particular initial conditions for which their differential equations were solved. It gave insight to the behavior of the model but the clear link to the experimental data was missing.
Line 79: what is RsbW, RsbV, RsbP and RsbU?
RE: This are the components of the SigB regulatory network used throughout the manuscript.
I don’t see why the author cites previous publications in his finding, is it a review? Can the author explain how he came to these equations?
RE: I cite the publication from which the chemical equations describing the system were adopted. The differential equations are original for this paper.
It is difficult for me to figure out the difference between previous data and this work findings.
RE: it is explained in the response to previous comment and it is also discussed in the Conclusions.
Eg: table 1: why does the author have to paste this table from ref 9 while the data were from ref 10? I am confused.
RE: Ref. 9 defines the chemical equations and corresponding rate constants, while Ref. 10 provided the kinetic data.

Reviewer 2 Report
The manuscript by Vohradsky et al reported a computer simulation of the SigB regulatory network in B. subtilis. The kinetic model of the SigB regulatory network was constructed using the well-known knowledge in the textbook, the kinetic parameters of the previous study, and the gene expression data available from the previous study. The author claimed that the free SigE level in the non-stressed condition should be very low and the kinetic parameters of reaction 5 (ch5) are more sensitive than that of reaction 6. Unfortunately, this study lacks the novelty and impact required for the IJMS as follows:
-
The novelty and biological significance of the simulation results are unclear. Low free SigE level in the non-stressed condition is well known as mentioned in the introduction. The parameter sensitivity was not performed comprehensively. The biological significance of the high sensitivity of the free sigB level on the kinetic parameters of reaction 5 was not discussed in the biological context.
-
Poor Materials and method. The section lacks many many essential descriptions of the simulation procedure used in this study such as the software, DOE solver, and setting used for the simulation. Which function or parameters were used for the data smoothing? Moreover, the text lacks an explanation of the input of the simulation model.
-
The model formulation is confusing. For example, it is unclear whether W2 in (a2) and W2f in (a5) are identical.
-
Confusing description. For example in the Abstract, “total amount of free sigma B is relatively high in the unstressed population, the amount of free SigB is quite low”. It is very confusing.
Author Response
The manuscript by Vohradsky et al reported a computer simulation of the SigB regulatory network in B. subtilis. The kinetic model of the SigB regulatory network was constructed using the well-known knowledge in the textbook, the kinetic parameters of the previous study, and the gene expression data available from the previous study. The author claimed that the free SigE level in the non-stressed condition should be very low and the kinetic parameters of reaction 5 (ch5) are more sensitive than that of reaction 6. Unfortunately, this study lacks the novelty and impact required for the IJMS as follows:
- The novelty and biological significance of the simulation results are unclear. Low free SigE level in the non-stressed condition is well known as mentioned in the introduction. The parameter sensitivity was not performed comprehensively. The biological significance of the high sensitivity of the free sigB level on the kinetic parameters of reaction 5 was not discussed in the biological context.
RE: I suppose that SigE means SigB. There is a contradiction. In the original paper by Keijser et al. the surprisingly high levels of SigB was found and discussed with no explanation. The low levels of free SigB has been expected but not quantified. But mainly I see the contribution of this study in defining precisely the kinetics of all the components of the system, either free molecules or complexes under conditions given by the Keijsers experiment, i.e. in natural conditions of growths. Also the effect of variation of kinetic constants and phosphatases amounts was presented. This provides insight into how the whole SigB regulatory system works. That was not made before and in my opinion it is important to have this knowledge.
- Poor Materials and method. The section lacks many essential descriptions of the simulation procedure used in this study such as the software, DOE solver, and setting used for the simulation. Which function or parameters were used for the data smoothing? Moreover, the text lacks an explanation of the input of the simulation model.
RE: changes to the Methods section concerning the simulation parameters were made and the paragraph was rewritten in this part.
The following sentences were added to the Methods:
"Since the expression data is noisy, the data were interpolated using Matlab function interp1 (method ‘pchip’) at 1-minute time steps and smoothed (Matlab function smoothdata, with a Gaussian-weighted moving average filter - method ‘gaussian’) prior to computation.
The expression profiles used for the computations are available in Supplementary file 1.The system of differential equations was solved using Matlab ode23 solver with initial values set to zero."
- The model formulation is confusing. For example, it is unclear whether W2 in (a2) and W2f in (a5) are identical.
RE: Sincere apologies. After careful checking of the equations a1-a10 I found this and other errors caused by misspelling in the equation editor. All equations were rewritten and checked for possible mistakes.
- Confusing description. For example in the Abstract, “total amount of free sigma B is relatively high in the unstressed population, the amount of free SigB is quite low”. It is very confusing.
RE: Yes. There was a mistake in the sentence, it should have been “total amount of sigma B is relatively high in the unstressed population, the amount of free SigB is quite low”.
Changes were made.
